# Recombinant Simian Varicella Virus-Simian Immunodeficiency Virus Vaccine Induces T and B Cell Functions and Provides Partial Protection against Repeated Mucosal SIV Challenges in Rhesus Macaques

**DOI:** 10.3390/v14122819

**Published:** 2022-12-17

**Authors:** Bapi Pahar, Wayne Gray, Marissa Fahlberg, Brooke Grasperge, Meredith Hunter, Arpita Das, Christopher Mabee, Pyone Pyone Aye, Faith Schiro, Krystle Hensley, Aneeka Ratnayake, Kelly Goff, Celia LaBranche, Xiaoying Shen, Georgia D. Tomaras, C. Todd DeMarco, David Montefiori, Patricia Kissinger, Preston A. Marx, Vicki Traina-Dorge

**Affiliations:** 1Division of Comparative Pathology, Tulane National Primate Research Center, Covington, LA 70433, USA; 2School of Medicine, Tulane University, New Orleans, LA 70118, USA; 3Biology Department, University of Mississippi, Oxford, MS 38677, USA; 4Division of Immunology, Tulane National Primate Research Center, Covington, LA 70433, USA; 5Division of Veterinary Medicine, Tulane National Primate Research Center, Covington, LA 70433, USA; 6Division of Microbiology, Tulane National Primate Research Center, Covington, LA 70433, USA; 7Department of Epidemiology, School of Public Health and Tropical Medicine, Tulane University, New Orleans, LA 70118, USA; 8Division of Surgical Sciences, Department of Surgery, Duke University School of Medicine, Durham, NC 27710, USA; 9Department of Tropical Medicine, School of Public Health and Tropical Medicine, Tulane University, New Orleans, LA 70118, USA

**Keywords:** rhesus macaque, vaccine, varicella virus-vector, SIV, protection

## Abstract

HIV vaccine mediated efficacy, using an expanded live attenuated recombinant varicella virus-vectored SIV rSVV-SIVgag/env vaccine prime with adjuvanted SIV-Env and SIV-Gag protein boosts, was evaluated in a female rhesus macaques (RM) model against repeated intravaginal SIV challenges. Vaccination induced anti-SIV IgG responses and neutralizing antibodies were found in all vaccinated RMs. Three of the eight vaccinated RM remained uninfected (vaccinated and protected, VP) after 13 repeated challenges with the pathogenic SIVmac251-CX-1. The remaining five vaccinated and infected (VI) macaques had significantly reduced plasma viral loads compared with the infected controls (IC). A significant increase in systemic central memory CD4+ T cells and mucosal CD8+ effector memory T-cell responses was detected in vaccinated RMs compared to controls. Variability in lymph node SIV-Gag and Env specific CD4+ and CD8+ T cell cytokine responses were detected in the VI RMs while all three VP RMs had more durable cytokine responses following vaccination and prior to challenge. VI RMs demonstrated predominately SIV-specific monofunctional cytokine responses while the VP RMs generated polyfunctional cytokine responses. This study demonstrates that varicella virus-vectored SIV vaccination with protein boosts induces a 37.5% efficacy rate against pathogenic SIV challenge by generating mucosal memory, virus specific neutralizing antibodies, binding antibodies, and polyfunctional T-cell responses.

## 1. Introduction

Almost four decades have passed since the human immunodeficiency virus (HIV) was first identified as the etiological agent of AIDS, with many successful antiretroviral therapies (ART) having been developed and in use. However, no effective prophylactic vaccine has been developed and HIV continues to spread. In 2020, UNAIDS estimated that 37.7 million adults and children were infected with HIV/AIDS worldwide and that there were 1.5 million newly acquired cases [1]. There were 650,000 people that died of AIDS related causes in the year 2021 alone which amounts to one death every minute (https://aidsinfo.unaids.org/). The recent recognition of a more pathogenic variant of HIV-1 circulating in the Netherlands places increased pressure on the need for an effective prophylactic vaccine [2]. Prior to the COVID-19 pandemic, the availability of healthcare and multiple antiretroviral drugs for many in developed countries significantly reduced HIV transmission, viral loads, and disease progression, resulting in an extension of life for many HIV infected persons. Despite these improvements, 25% of HIV infected individuals live in lower resourced countries where therapy availability is minimal and often unavailable. Even in the year 2021, an estimated 800,000 children living with HIV were found that were not being treated with ART [3]. During the COVID-19 pandemic, health care and supply chains for ART have been severely disrupted and there has been increased incidence of new infections. HIV infection was also shown to be associated with more adverse COVID-19 outcomes [4] and an increased risk of mortality [5,6]. Therefore, the development of an effective HIV vaccine is urgently needed to completely control this pandemic by the year 2030.

Varicella zoster virus (VZV) infection causes varicella (chickenpox) in children, and herpes zoster (shingles), primarily in the elderly. The live, attenuated VZV Oka vaccine is safe and effective for prevention of childhood chickenpox and is recommended for children in several countries [7]. It has been shown to produce strong humoral and CD8 T-cell responses [8] and has a greater than 97% efficacy against severe disease [9]. The vaccine also prevents herpes zoster and post-herpetic neuralgia in the elderly [6]. In addition, the varicella vaccine is safe and effective in groups of immunosuppressed patients that have been tested, including children with HIV infection [10,11,12]. Thus, aVZV-based vaccine is also an attractive candidate as a recombinant vaccine vector expressing the antigens of other pathogens [13]. Recombinant VZV vaccines express hepatitis B surface antigens, HIV-1 gp120, and mumps virus (MuV) hemagglutinin-neuraminidase (HN) protein-induced immune responses to the foreign antigens in immunized guinea pigs [14,15,16]; a rVZV Oka vaccine expresses herpes simplex type 2 (HSV-2) glycoproteins B and D and induces neutralizing antibodies and immune protection against HSV-2 challenge in immunized guinea pigs [17,18]. An earlier study by Matsuura et al. also demonstrated that recombinant VZV vaccines expressing the MuV HN and fusion (F) genes were able to produce immunogenicity and the production of HN and F neutralizing antibodies in guinea pigs [19]. The authors stated that the recombinant VZV can be a promising vaccine candidate for polyvalent protection against both VZV and MuV [19]. However, stringent human host range restrictions limit the development of an experimental animal model for the efficacy testing of candidate varicella vaccines.

The simian varicella model has provided an experimental approach in the investigation of varicella pathogenesis and the evaluation of antiviral agents and vaccines [20,21]. Simian varicella virus (SVV) is closely related to VZV and causes a natural, varicella-like disease in nonhuman primates (NHP) [20,22,23]. Construction of recombinant SVV (rSVV) vaccines expressing simian immunodeficiency virus (SIV) antigens and testing in the simian AIDS model offers an important new experimental approach in the evaluation of SIV/HIV vaccine candidates. Our previous studies describe construction of rSVV vaccines that expressed SIV antigens and successfully induced SIV immune responses in immunized African green monkeys [24]. Vaccination with rSVV-live attenuated rSVV/SIVgag and rSVV/SIVenv vaccines effectively induced humoral and cellular mucosal immune responses to SIV antigens in immunized rhesus macaques (RMs), and significantly reduced SIV viral loads following intravenous challenge with pathogenic SIVmac251-CX-1 [25] and increased CD4+ T cell proliferation, correlating with reduced plasma viral load in SIV challenged rSVV-SIV vaccinated RMs [26].

In this study, we extend those findings with subcutaneous and intranasal vaccinations to induce both systemic and mucosal immune responses with live, attenuated rSVV vaccinations followed by SIV-Gag and SIV-Env protein boosts in female RMs. We demonstrate that vaccine regime significantly enhanced mucosal memory T cell population, induced SIV neutralizing antibodies, antigen specific V1/V2 binding antibodies, and antigen-specific polyfunctional T cell responses, reduced SIV plasma viral loads, and yielded 37.5% efficacy of protection following multiple intravaginal pathogenic SIVmac251-CX-1 challenges.

## 2. Materials and Methods

### 2.1. Ethics Statement

SIV infection of RM is the most widely used NHP model of HIV/AIDS and is also widely used for testing vaccines and cure strategies [27]. RMs were bred at Covance Research Products, Alice, TX and transported to the Tulane National Primate Research Center (TNPRC) for this study. All animal housing, care, and research were performed in compliance with *The Guide for the Care and Use of Laboratory Animals* (National Research Council) and the guidelines at the TNPRC, an institute fully accredited by the Association of the Assessment and Accreditation of Laboratory Animal Care and in accordance with the Animal Welfare Act guidelines. Animal experiments were reviewed and approved by the Institutional Animal Care and Use Committee of Tulane University (protocol P0451).

Animals were singly housed indoors in climate-controlled conditions with a 12/12-light/dark cycle. Caging was equipped with perches and several manipulatable objects that exceeded the USDA standards for minimum cage sizes. The animals were fed commercially prepared NHP diet twice daily, supplemented by varied feeding enrichment. Water was available ad libitum through an automatic watering system. All subjects were monitored twice daily to ensure their welfare. Any abnormalities, including those of appetite, stool, and behavior, were recorded and reported to a veterinarian. The TNPRC has an IACUC-approved Environmental Enhancement Program (EEP) to promote psychological wellbeing. Program components include social housing where compatible with research objectives, physical enrichment to promote species-appropriate exploratory and locomotor behaviors, as well as to provide foraging opportunities and novelty. Welfare is closely monitored, and signs of impaired wellbeing are assessed by behavioral management personnel to quantify the behavior and identify triggers. Based on this information, personalized interventions are implemented, and may include additional devices such as puzzles, targeted human interaction or positive reinforcement training, alterations in the social setting, and/or more frequent and varied feeding enrichment. Veterinarians in the TNPRC have established procedures to minimize pain and distress using several approaches. For vaccinations and other procedures, animals were anesthetized with ketamine-HCl (10 mg/kg) or tiletamine/zolazepam (8 mg/kg) by intramuscular (IM) route and given buprenorphine (0.01 mg/kg IM) for pain. This method is consistent with the recommendation of the American Veterinary Medical Association Guidelines on Euthanasia (https://www.avma.org/KB/Policies/Pages/Euthanasia-Guidelines.aspx 4 November 2022), with administration of buprenorphine (0.01 mg/kg) followed by an overdose of pentobarbital sodium. Death was confirmed by loss of heartbeat and corneal reflex, and dilation of the pupils.

### 2.2. Cells and Virus

Vero (African green monkey kidney) cells were grown in Eagle’s Minimal Essential Medium (EMEM) supplemented with 5% fetal bovine serum (Atlanta Biologicals, Atlanta, GA), penicillin (5000 U/mL), and streptomycin (5000 U/mL). The SVV Delta herpesvirus strain, was originally isolated from a Patas monkey at the Delta Regional Primate Research Center, now the TNPRC [28]. SIVmac251-CX-1, a cryopreserved viral stock obtained from the Virus Characterization, Isolation, Production and Sequencing Core Division of microbiology at TNPRC, and previously named and referenced as SIVmac251-PM, was utilized for repeated intravaginal (IVAG) challenge [29].

### 2.3. Recombinant Vaccine Design

The recombinant, live, attenuated SVV vaccine vector expressing either the *env* and *gag* genes of SIVmac239 (rSVV-SIV*env/gag*) or the glycoprotein G gene of Respiratory syncytial virus (RSV*_G_*), under the control of the human cytomegalovirus (CMV) immediate early promoter/enhancer, was constructed using the SVV cosmid recombination system, as previously described [24,30]. Expression of the SIV and RSV antigens in infected Vero cells was confirmed by immunoblot assay prior to immunization.

### 2.4. Animal Cohort

Fifteen female, Indian-origin RMs (*Macaca mulatta*), 4–6 years old and weighing 4.6–6.0 kg, were shown seronegative for SVV, SRV and SIV (Appendix A). Female macaques were utilized to test the IVAG route of challenge and to mimic one of the major routes of HIV transmission in humans. All chosen RMs were negative for known SIV elite controlling alleles of *Mamu-A*01*, *Mamu-*B**08* and *B*17* based on PCR assay [31]. Upon arrival at TNPRC, macaques were held for a 90-day quarantine and shown to be healthy by physical exams and blood tests. The RMs were divided into two groups (vaccine and control), with eight and seven macaques, respectively.

### 2.5. rSVV-SIV Immunization and SIV Challenge

At the start of the immunization regime, on day 0, the vaccine group of animals received both subcutaneous (SC) and intranasal (IN) inoculations of 1.5 × 10^6^ pfu of rSVV-SIV239*gag* and 1.5 × 10^6^ pfu rSVV-SIV239*env* infected Vero cells. Control group also received similar immunizations with 1.5 × 10^6^ pfu of rSVV-RSV*_G_*-infected Vero cells, containing the RSV G gene. Booster immunizations with the same vaccines were administered on weeks 8, 24, 43 and 54 wpi (weeks post initial immunization) to total five immunizations (Figure 1A). For the vaccine RMs, an additional 1 mL booster immunization of 100 µg purified SIVmac251 pr55 Gag protein (NIH HIV Reagent Program, Rockville, MD, USA) and SIV/M766 gD-gp120 Env protein (kind gift of Steve Whitney, ABL, Inc., Rockville, MD, USA) with 1x Adjuplex adjuvant (Emperion, Inc., Columbus, OH, USA) was administered by IM in the upper thigh on immunization weeks 24, 43 and 54 wpi, for vaccinations #3, #4, and #5, while the control animals were similarly administered a 1 mL PBS/Adjuplex mock vaccine for the same vaccine time points. Six weeks after the final 5th immunization, all RMs were IVAG challenged, 1mL per week for up to 13 weeks, with repeated, escalating dose 316–1265 TCID_50_ per mL of pathogenic SIVmac251-CX-1. This is the low passage cryopreserved live SIV stock generated in CEMx174 from an early passage rhesus macaque-derived SIVmac251-infected HuT78 virus stock, comprising a virus swarm that was obtained from Dr. Richard Heberling of Southwest Foundation for Biomedical Research to coauthor Dr. Preston Marx [32] and was also used in our previous study [25]. Weekly challenge doses were 316 TCID_50_ (Weeks 1–4); 632 TCID_50_ (Weeks 5–11); and 1,265 TCID_50_ (Weeks 12 & 13) for a total 13 weeks to assure all control macaques became infected. RMs were given weekly physical exams and bled post challenge to determine SIV infection and/or clinical signs of SIV infection.

### 2.6. Clinical Evaluations

Macaques were monitored with monthly physical examinations and clinical hematological and chemical analyses for signs of well-being and infection. Parameters of acute SVV virus infection were evaluated as previously described [28,33]. Any evidence of typical lesions of varicella rash were regularly evaluated, especially following each vaccination. SVV viremia was monitored by harvesting peripheral blood mononuclear cells (PBMCs) and co-culture on Vero cell monolayers. Liver function was assessed by measuring liver enzymes, aspartate transaminase (AST), alanine aminotransferase (ALT), and lactate dehydrogenase (LDH) in blood samples. RMs were given physical examinations with blood and tissue sampling bimonthly during the immunization phase and once or twice per week during the challenge phase. Any macaques becoming SIV infected were monitored up to twelve weeks post SIV infection. All subjects were euthanized according to TNPRC SOP 3.23 and multiple tissues were harvested for further analyis.

### 2.7. Peripheral Blood: Lymph Node (LN) and Intestinal Cells Processing Techniques

PBMCs were isolated from Na heparin-anticoagulated whole blood by Ficoll-Hypaque density gradient centrifugation (GE Healthcare, Chicago, IL, USA) as described elsewhere [34]. Peripheral LNs were processed as reported earlier [35]. Jejunum and rectal pinch biopsies were collected in complete RPMI media and transferred to laboratory in ice cold condition. Lamina propria lymphocytes (LPLs) were isolated using previously described protocols [34,36,37]. All cells were washed in PBS and finally resuspended in complete RPMI-1640 medium containing 10% fetal bovine serum (FBS) before staining. Cells were counted using the trypan blue dye exclusion method and were found >90% viable during staining.

### 2.8. Quantitative SIV Plasma Virus Load (VL)

SIV RNA copies per milliliter were determined by a two-step real-time qPCR assay at NHP Core Virology Laboratory at Duke University using frozen EDTA plasma samples. A QIAsymphony SP (Qiagen, Hilden, Germany) automated sample preparation platform along with a Virus/Pathogen DSP midi kit and the *cellfree500* protocol was used to extract viral RNA from plasma. A reverse primer specific to the gag gene of SIVmac251 (5′-CAC TAG GTG TCT CTG CAC TAT CTG TTT TG-3′) was annealed to the extracted RNA and then reverse transcribed into cDNA using SuperScript^TM^ III Reverse Transcriptase (Thermo Fisher Scientific, Waltham, MA, USA) along with RNAse Out (Thermo Fisher Scientific). The resulting cDNA was treated with RNase H (Thermo Fisher Scientific) and then added (2 replicates) to a custom 4x TaqMan™ Gene Expression Master Mix (Thermo Fisher Scientific) containing primers and a fluorescently labeled hydrolysis probe specific for the gag gene of SIVmac251 (forward primer 5′-GTC TGC GTC ATC TGG TGC ATT C -3′, reverse primer 5′- CAC TAG GTG TCT CTG CAC TAT CTG TTT TG-3′, probe 5′-/56-FAM/CTT CCT CAG TGT GTT TCA CTT TCT CTT CTG CG/3BHQ_1/-3′). The qPCR was then carried out on a StepOnePlus™ Real-Time PCR System (Thermo Fisher Scientific) using the following thermal cycler parameters: 50 °C for 2 min, 95 °C for 10 min, then the following parameters are repeated for 50 cycles: 95 °C for 15 s, cool to 60 °C for 1 min. Mean SIV *gag* RNA copies per reaction were interpolated using quantification cycle data and a serial dilution of a highly characterized custom RNA transcript containing a 730 bp sequence of the SIV *gag* gene. Mean RNA copies/mL were determined by applying the assay dilution factor (DF = 18.72). The limit of quantification (LOQ) for this assay is 62 RNA copies /mL of plasma [38].

### 2.9. Quantification of SIV Antigen-Specific IgG Responses

SIV-Gag and Env-specific IgG were detected in plasma using ELISA as previously described [39]. In brief, purified SIV Gag or Env antigens (Immunodx, Woburn, MA, USA) were used as coating antigen. After washing, serial four-fold dilutions of plasma in blocking buffer were added in wells. Following washing, plates were further incubated with peroxidase-conjugated goat anti-monkey IgG secondary antibodies (Exalpha Biologicals, Shirley, MA, USA) diluted in blocking buffer. After washing, all wells were further incubated with O-Phenylenediamine Dihydrochloride substrate (Sigma-Aldrich, St. Louis, MO, USA), color was developed, and reaction was stopped by adding 4N sulfuric acid. The plate was read at an optical density (OD) of 490nm using Elx800 reader (Biotek, Winooski, VT, USA). All samples were assayed in duplicates with appropriate positive and negative controls. For quantification of SIVGag or Env specific IgG responses, rhesus IgG (NIH-NHP Reagent Resource) standard was used. Nonlinear regression using a sigmoidal dose-response variable slope model was used to interpolate concentrations from the standard curve as reported earlier [39].

### 2.10. Quantification of SIV_MAC_ Neutralization Antibody Titers

Neutralizing antibody (Nab) activity was measured in 96-well culture plates by using Tat-regulated luciferase (Luc) reporter gene expression to quantify reductions in virus infection in TZM-bl cells [40]. TZM-bl cells were obtained from the NIH AIDS Research and Reference Reagent Program, as contributed by John Kappes and Xiaoyun Wu. Assays were performed with replication-competent SIVmac251-CX-1 virus stocks or SIV Env-pseudotyped viruses as described previously [40]. Serum samples were heat-inactivated at 56 °C for 30 min prior to assay. Test samples were diluted over a range of 1:20 to 1:43,740 in cell culture medium and pre-incubated with virus (~150,000 relative light unit equivalents) for 1 hr at 37°C before addition of cells. For some viruses, samples were diluted, 1:30 to 1:2,343,750 or 1:300 to 23,437,500 in order to achieve an end-point titer. Following a 48-hr incubation, cells were lysed and Luc activity determined using a microtiter plate luminometer and BriteLite Plus Reagent (Perkin Elmer). Neutralization titers are the sample dilution (for serum) or concentration (for monoclonal antibodies) at which relative luminescence units (RLU) are reduced by 50% compared to RLU in virus control wells after background subtraction of RLU in cell control wells.

### 2.11. Quantification of SVV Vector Antibody Titers

Humoral immune responses to the SVV vector were determined by an SVV ELISA assay [41]. SVV antigen, prepared from lysates of SVV infected Vero cells, was coated to 96 well ELISA plates (500 ng/well). Dilutions of serum samples were added to wells for one hr at 25 °C. After washing, mouse anti-monkey IgG horseradish peroxidase conjugated antibodies was added for one hr at 25 °C, followed by washing and incubation with TMB (3,3′,5,5’ tetramethylbenzidine) substrate for 20 min. After addition of stop buffer, sample absorbance at OD of 450 nm was determined on an ELISA reader [42]. SVV antibody titers were expressed as the reciprocal of the serum dilution that resulted in 450 nm absorbance readings above the determined cut-off value.

### 2.12. Binding Antibody Multiplex Assay (BAMA)

Macaque SIV-Gag and SIV-Env-specific IgG binding antibodies in serum were measured with a custom SIV binding multiplex antibody assay (SIV-BAMA), as previously described [43] with a panel of SIV-Gag, SIV-Env antigens and SIV V1/V2 gp70 scaffold antigens. Serum samples were tested with 5-fold serial dilutions starting at 1:80 for a total of 6 dilutions, and the binding magnitude was reported as Area-Under-the-Curve (AUC). Vaginal secretions were eluted off the vaginal sponges, filtered and re-concentrated for serological assays. Total IgG concentrations of prepared vaginal samples were determined by an in-house ELISA. Magnitude of SIV antigen-specific binding for vaginal samples is defined as Specific Activity (SA), which is calculated as mean fluorescence intensity (MFI) after blank/MulVgp70 subtraction × Dilution / total IgG concentration (µg/mL). Serum positivity criteria (determined at a dilution of 1:80) were as follows: (i) MFI after blank/MulVgp70 subtraction is at least 100, (ii) MFI after blank/MulVgp70 subtraction higher than the Ag-specific cutoff (95^th^ percentile of all baseline binding per antigen), and (iii) MFI 3-fold higher than that of the matched baseline before and after blank/MulVgp70 subtraction. Vaginal samples’ positivity criteria were as follows: (i) MFI after blank/MulVgp70 subtraction is at least 100, (ii) SA 3-fold higher than that of the matched baseline, and (iii) SA higher than the Ag-specific cutoff (95th percentile of all baseline SA). All BAMAs were performed in a blind fashion using polystyrene beads, and the subtraction of blank/MulVgp70 MFI was to control for non-specific bead binding.

### 2.13. Tissue T-cell Immunophenotyping

Lymphocytes isolated from peripheral blood, LN, jejunum, and rectum were quantified for their activation (CD38+, CD69+, and HLADR+), proliferation (Ki67+), and naïve/memory phenotype (CD28+ and CD95+). Cells were first stained with live/dead stain (Thermo Fisher Scientific, Eugene, OR, USA). After washing, cells were further stained with cocktail of monoclonal antibodies (mAbs) including anti-CD3, anti-CD4, anti-CD8, anti-CD28, anti-CD95, anti-CD38, anti-CD69, and anti-HLADR antibodies (BD Biosciences, San Jose, CA, USA; Appendix A) as reported earlier. To detect proliferating Ki67+ cells, intracellular staining protocol was performed using BD Fix/Perm solutions as described earlier [26]. Cells were fixed with 1x BD stabilizing and fixative buffer. At least 20,000 events were collected by gating on lymphocytes from each sample, and the data were analyzed using FlowJo software, version 10.7.0. (Becton, Dickinson & Company, Franklin Lakes, NJ, USA).

### 2.14. Antigen-Specific Intracellular Cytokine Flow Cytometry Staining and Quantification

To assess the magnitude and functional characteristics of SIV-specific CD4+ and CD8+ T-cells from both groups of macaques, intracellular cytokine flow cytometry assay was performed using freshly isolated PBMC, LN lymphocytes and mucosal LPLs, as described previously [26,36,44]. Briefly, isolated cells were resuspended in complete RPMI media and stimulated with SIV-Gag (NIH HIV Reagent Program, Cat. No. 6204) and SIV-Env (NIH HIV Reagent Program, Cat. No. 6883) in presence of 0.5µg/mL of anti-CD28, anti-CD49d mAbs (BD Biosciences). For degranulation marker staining, anti-CD107a and anti-CD107b mAbs were added at the time of cell stimulation (Appendix A). After stimulation for 6 hrs in presence of Brefeldin A (Sigma) and Monensin (BD Biosciences), cells were washed and stained with live/dead, then surface staining with anti-CD3, anti-CD4, anti-CD8, and anti-CD45 mAbs (Appendix A). Cells were permeabilized and further stained with intracellular staining with anti-IL2, anti-IL4, anti-IL5, anti-IL21, anti-IFNγ, anti-TNFα mAbs (Appendix A). Cells were fixed with 1x BD stabilizing and fixative buffer. Data were acquired using a BD Fortessa instrument. Cells were gated on singlets, live cells, followed by CD45+ cells, and then on CD3+ T-cells and subsequently on CD3+CD4+ and CD3+CD8+ T-cell subsets. Each subset of T cells was further analyzed for the presence of cytokines using FlowJo software. Boolean combination gates were used to define polyfunctional cytokine positive cells. The criterion for a positive cytokine response was greater than 0.05% and a two-fold increase in the frequency for that specific antigen and cytokine above the medium control culture. All positive values were subtracted from the values of medium control.

### 2.15. Statistical Analyses

Statistical analyses for both neutralizing and binding antibody data were conducted using Repeat Measures T-test analyses. The first analysis of SIV viral load was a Repeat Measures ANOVA, using Proc mixed with missing data imputed using Proc MI in SAS. The purpose of this analysis was to examine if the log-transformed mean viral load was higher among unvaccinated animals compared to those who were vaccinated. The second analysis of SIV viral load was time-to-event analysis, using Cox Proportional Hazard analysis. At day 1, all animals were infection free and were followed for SIV seroconversion. Time was calculated in weeks from the time they entered the study to seroconversion for up to 13 weeks. Animals who had not seroconverted by 13 weeks were censored and included for the length of the study. All analyses were conducted using SAS statistical software version 14.1. (SAS Institute, Cary, NC, USA). Infection hazard curves were also performed using Kaplan Meyer approach. One-way ANOVA was used to observe any statistically significant differences in different memory T cells and other cell populations between two groups. Bonferroni and Tukey–Kramer’s multiple comparison tests were applied for equal and unequal sample size, respectively, to identify statistically significant differences between the groups. Correlation analyses were performed assessing the association between peak viral load levels, and multiple cellular markers, using a Pearson’s correlation measurement. For uninfected animals, null values were used for peak viral load. 95% confidence intervals and statistical significances were calculated based on results using Stata 15 (StataCorp LLC. College Station, TX, USA). Graphical presentation of all immunological data were performed using GraphPad Prism version 9 (GraphPad Software, San Diego, CA, USA).

## 3. Results

### 3.1. Immunization and SIV Challenge

All the eight vaccine RMs received a total of five immunizations rSVV-SIV239*gag* and rSVV-SIV239*env* along with three gp120 and Gag protein boosters and Adjuplex. Seven control macaques received rSVV-RSV*_G_* immunogen five times along with PBS and Adjuplex for the last three times of immunization (Figure 1A). All immunizations were well tolerated, and macaques remained clinically normal throughout the period of the vaccination phase (see Appendix A).

All RMs were IVAG repeated and challenged with pathogenic SIVmac251-CX-1 6 weeks after the final immunization (Figure 1A). Seven and ten days following each of the 13 SIV challenges, the plasma VL was assessed and showed SIV presence and infection in all control RMs (infected controls, IC) (Figure 1B). Six of the seven control macaques were infected following one to eight challenges and had expected high VL, ranging from 1.3 × 10^6^–5.5 × 10^7^ viral copies/mL plasma. The last control, LK53, became infected following 13 challenges and had a slightly lower 5.0 × 10^5^ peak VL compared to the other control animals. For the vaccinated RMs, five of the eight became infected (vaccinated infected, VI), following two to seven challenges. However, of these five, only one vaccine animal, LK54, showed a high peak VL (9.1 × 10^6^ viral copies/mL) while the remaining four VI macaques (LK45, LK56, LK59, LK60) had lowered peak VL ranging from 9.2 × 10^3^–9.8 × 10^5^, compared to controls. The rest of the three vaccinated RMs, LK50, LK57, LK58, remained uninfected (vaccinated protected, VP) after 13 SIV challenges (Figure 1C). Mean plasma VL values calculated for each group, demonstrated a significant >2 log reduction in mean peak VL of the vaccine compared with control animals (Figure 1D), a reduction that persisted throughout the set point and monitoring period (*p* = 0.024 Repeat Measure ANOVA). An infection curve generated following each challenge showed a clear difference between the vaccinated and control groups, demonstrating the presence of three of eight (37.5%) VP macaques following the 13 SIV challenges, although the difference was not statistically significant (*p* = 0.120) (Figure 1E). This protection and significant >2 log reduction in plasma viremia of the VI macaques, however, was suggestive of the increased vaccine efficacy compared to our earlier study, probably due to the higher amount of antigen delivery in each vaccination, protein boost, and the duration of vaccination that might help to generate long-term mucosal effector memory T cells [25].

### 3.2. All Vaccinated Macaques Generated Strong Antigen-Specific IgG Responses after Intramuscular Protein Boosts

IgG responses in the vaccinated RMs to SIV p27 Gag and gp120 Env antigens were initially low but detectable (mean ± the standard errors = 668 ± 556 ng/mL and 147 ± 58 ng/mL, respectively) following the first two immunizations (Figure 2A,B). Env-specific responses at 24 weeks (post second immunization) trended higher in two VP macaques, LK50 and LK57 (4838 and 472 ng/mL of plasma for LK50 and LK57, respectively, Figure 2A) compared to the rest of the vaccinated macaques. Increased Env-specific IgG responses in all vaccine subjects from the third immunization onwards (mean ± the standard errors = 9816 ± 4411, 168,279 ± 96,518, and 185,513 ± 30,140 ng/mL) for 27 wpi, 45 wpi and the 58 and 60 wpi pre-challenge time points, respectively) demonstrated that vaccination followed by the protein boost was able to generate strong Env-specific IgG responses (Figure 2A).

The addition of SIV p27 Gag proteins in Adjuplex adjuvant for the third to fifth immunizations also showed 1–3 log increases in Gag specific IgG responses (mean ± the standard errors = 39,315 ± 18,279, 203,325 ± 152,826, and 335,493 ± 74,523 ng/mL) for 27 wpi 45 wpi, and 58 wpi pre-challenge time points, respectively, demonstrating that vaccination followed by protein boost was also able to generate strong Gag-specific antibody responses (Figure 2B). Similar to the Env-specific responses, LK50 and LK57 had increased SIV-Gag-specific IgG responses detected as early as the time point 27 wpi, two weeks following vaccination #3, with max levels at 111,568 ng/mL compared with the next highest responses of LK59, a VI macaque with 25,116 ng/mL and other points below. Following vaccination #4, the remaining RMs, including the third VP subject LK58, showed maximal increases with a range from 27,966 to 1,272,285 ng/mL by the vaccination #5 time point. The mean Gag-specific antibody response at the post-challenge time point was significantly higher than at any other time point in the study by ANOVA analysis (*p* < 0.05). Increased SIV-Gag and Env-specific binding IgG responses after protein boost might be the effect of initial recombinant vaccine-mediated B cell priming in vaccinated macaques.

The vaccinated RMs showed peak Env-specific IgG responses at 58 wpi, the end of the vaccinations phase, whereas peak Gag-specific IgG responses were detected and peaked earlier from 45–58 wpi. Of the five VI RMs, they all showed modest anamnestic Env-specific responses after SIV infection and only two of the five showed any Gag-specific anamnestic response. For the VP macaques, they all showed a decline in Env-specific antibody levels, and two of the three showed some decline in Gag-specific responses, indicating a lack of anamnestic response in the VP group which is consistent with their uninfected status.

### 3.3. Neutralizing Antibodies to the SIVmac-CX-1 Challenge Virus was Elicited following the Serial Vaccine Regime

The ability of the vaccine to elicit Nabs against the SIVmac251-CX-1 challenge virus was evaluated at time points: pre, 58 wpi (pre-challenge) and 1 month post challenge and confirmed SIV infection. After the fifth immunization at 58 wpi, all eight vaccinated macaques demonstrated detectable Nab titers ranging from 90,611 to 1,897,415 ID_50_. Two VP macaques, LK50 and LK57, demonstrated increased Nab titers with 1,829,690 and 1,897,415 ID_50_, respectively, compared to VI subjects at the 58 wpi time point. The third VP macaque, LK58, had the lowest Nab titer at the 58 wpi time point (90,611 ID_50_, Figure 2C). Conversely, following SIV challenge and infection, four of the VI SIV+ macaques (LK45, LK54, LK56, LK60) produced increased Nabs due to stimulation from circulating SIV antigens following SIV infection while the three VP macaques (LK50, LK57, LK58) showed reduced Nab titers of 171,217, 281,464, and 26,365 ID_50_, respectively (Figure 2C).

### 3.4. SVV Vector Antibody Responses following Vaccination

Mean SVV-specific IgG titers increased steadily following each of the first three vaccinations: pV#1 @10,500 (4000–16,000), pV#2 @17,500 (8000–32,000), and pV#3@ 23,000 (8000–32,000), respectively, and remained stable following the last two vaccinations, at pV#4 and pV#5 @23,000 (8000–32,000) for the vaccine group (Figure 2D). At one month post SIV infection, the mean SVV vector IgG titers were reduced to 10,000 (4000–32,000).

### 3.5. SIV-Gag-, SIV-Env- and gp70 V1/V2 Binding Antibodies Detected in Both VP and VI RMs

We were further interested in identifying vaccine specific non-neutralizing binding antibodies. Sera from both the 10 wpi and the 56 wpi vaccination time points were analyzed against a panel of SIV-Gag, SIV-Env antigens and SIV V1/V2 gp70 scaffold antigens from multiple SIVs: SIVmac239, SIVmac251 and SIV/E660 (Figure 3A,B). Induction of SIV-specific antibodies at 10 wpi detected both SIV Gag antigens p27 and p55, ranging from 2167–23,951 AUC for six of eight vaccinated RMs. Of note, the two highest p27 responses were in VI LK59 (7103), which resisted two SIV vaginal challenges, and VP LK50 (5297), which resisted all 13 SIV challenges. Low-level IgGs were detected against the gp140 SIVsmE660 antigen, ranging from 572–5103 AUC in five of eight vaccinated RMs with the highest responses for VI LK59 (5,103) and two VP subjects, LK57 and LK50 (3618, 2331). No systemic induction of IgG against SIVmac251 gp130 Env, SIVmac239 gp120 Env or gp70 V1/V2 antibodies were detected at the 10 wpi time point. Serum IgG production against p27 and p55 Gag antigens increased at 56 wpi (78,800–97,840 AUC). Increased IgG responses against gp130 SIVmac251 envelope (54,201–71,147 AUC) and gp120 SIVmac239 envelope (54,257–71,390), as well as gp70 V1/V2 SIVmac251 (38,128–79,777) and gp70 V1/V2 SIVmac239 (36,811–81,240), were observed in all vaccinated RMs compared to the 10 wpi time point. A V1V2-specific binding antibody response was identified as one of the correlates of protection in the RV144 Thai trial [45]. The three highest responses to the gp70 V1/V2 envelope epitopes were in LK45, the VI SIV+ infected RM that resisted seven challenges, and two of three VP macaques, LK57, LK50 for SIVmac239 (81,240, 71,281, and 61,528, AUC, respectively) and for SIVmac251 (79,777, 70,665, and 60,079 AUC, respectively, Figure 3A,B).

SIV-specific IgG binding antibodies were also evaluated in vaginal secretions from vaccinated macaques at 10 and 56 wpi against several SIV antigens. Levels of Gag antibodies for p27 ranged from 72,554 to 484,057 SA, and for p55 from 72,580 to 919,818 SA, with the VP LK50 having the highest p27 and p55 Gag responses, at 484,057 and 919,818 SA, respectively, at 10 wpi (C). The SIV envelope and gp70 V1/V2 antibody responses were slightly lower than Gag responses in the mucosal compartment, with most RMs ranging between 3,696–67,295 SA except for VI LK45, with values for gp120 SIVmac239 and gp130 SIVmac251Env (271,840 and 337,834 SA) and gp70 V1/V2 (332,832, 298,212, and 204,727 SA). At the 56 wpi time point, the antigen-specific IgG responses for both Gag p27 and p55 (607,513 and 795,826) and Env of gp130 SIVmac251 and gp140 SIVsmE660 (119,665 and 302,597, respectively) were higher in one VP macaque, LK57, compared to other vaccinated macaques (Figure 3C,D). Vaginal antibody data from LK50 at 56 wpi were not available. No significant differences in the binding antibody levels between all VI and VP subjects at either the 10 or 56 wpi time point was detected.

### 3.6. Impact of Vaccination on the Distribution of Naïve and Memory CD4+ and CD8+ T Cell Populations in PBMC and LN Lymphocytes

Dynamics of naïve (CD28+CD95-), central memory (CM, CD28+CD95+), and effector memory (EM, CD28-CD95+) CD4+ T cell populations were monitored from baseline through all vaccination time points for both control and vaccine groups. At 2 weeks post V#3 and V#4 time points, the CM CD4+ frequency in PBMC was significantly increased in the vaccine group compared to the controls (*p* < 0.05 and *p* < 0.01 for pV#3 and pV#4 time points, respectively) (Figure 4A). This increase of CM CD4+ T cell population was also detected at pV#3 compared to the pV#2 time point in PBMC in vaccinated macaques (Appendix A). No significant difference in the naïve and EM CD4+ population in PBMC was shown any time point between the vaccine and control groups (unpublished work). In LNs, a significant decrease in naïve CD4+ T cells at pV#4 was detected in vaccinated RMs when compared to controls (Figure 4B, *p* <0.001). In contrast, there were significant increases in CM (Figure 4C, *p* < 0.05) and EM (Figure 4D, *p* <0.01 to 0.001) CD4+ populations, detected following the pV#3 or pV#4 time points in vaccine macaques compared to controls. The CM population also increased at pV#5 in the three VP macaques, although the values were not statistically significant when compared to VI macaques. Significant increases in the CM CD4+ T cell population, however, were detected in the vaccinated macaques at the pV#5 time point compared to control groups, suggesting that the difference in vaccine regimen which includes the recombinant vectored vaccine along with the viral proteins boost had a major impact in inducing more CM population in LN tissues (Appendix A). In contrast, EM CD4+ T cell population in LN decreased in the vaccine macaques after vaccination, and the difference in frequency at pV#5 compared to the baseline (pre) and other vaccine time points was statistically significant (Appendix A). The dynamics of naïve, CM, and EM CD8+ T cell population in PBMC and LN tissues from vaccine and control macaques were also calculated, but none were significantly different between those two groups.

### 3.7. Impact of Vaccination on the Distribution of Total CD4, Naïve and Memory CD4+ and CD8+ T cell Populations in Mucosal Tissues

CD4+ T cells dynamics in jejunum and rectal LPL were measured. We did not observe any significant changes in CD4+ T cells frequency in either jejunum or rectal LPL at any pre challenge time points between vaccine and control macaques. However, after challenge there was a significant loss of CD4+ T cells both in jejunum and rectal LPL tissue compared to pre vaccine time points (Figure 5B,C, Appendix A). However, the VP subjects, LK50, LK57, and LK58, all maintained the mucosal CD4+ populations. No significant differences in CD4+ T cell population were detected in rectal CD4+ T cells at one month post SIV+ confirmed time point between the VI and IC groups. A significant decrease in rectal CD4+ T cell population was detected at the SIV+ time point in the VI group when compared to pre and other vaccination time points (Appendix A). In contrast, the jejunum CD4+ T cell population was significantly different between the vaccine and control groups at post challenge SIV+ time point (*p* < 0.05). This significant difference is likely due to the maintenance of CD4+ T cell population in these three VP macaques compared to the other five VI macaques at post challenge time point (Figure 5B). No significant differences were detected in jejunum or rectal naïve, CM, and EM CD4+ T cell populations during vaccination phase between vaccine and control macaques (Appendix A and Appendix A).

Importantly, the jejunum and rectal CM CD8+ T cell percentages were reduced significantly in all vaccine macaques starting from the post V#4 time points compared to the controls (*p* < 0.05 to 0.001, Figure 5A,D,F). A significant increase in jejunum CM CD8+ T cell population was detected in the vaccine group at pV#2 and pV#3 compared to pre time point (Appendix A). However, the frequency of jejunum CM CD8+ T cells decreased significantly at the pV#5 time point in all vaccine macaques (Appendix A). Conversely, a significant increase in the percentages of EM CD8+ T cells in the jejunum and rectal LPL starting from pV#4 was detected in the vaccine macaques compared to the control macaques (*p* < 0.05 to 0.001, Figure 5E,G). Pearson’s correlation analysis demonstrated a significant negative correlation between rectal EM CD8+ T cell frequencies at both pV#4 and pV#5 time points and peak plasma VL in RMs (*p* = 0.007 Figure 5H and *p* = 0.028, respectively). Jejunum EM CD8+ T cell frequencies increased significantly at pV#5 compared to pre time point in vaccine group (Table S3). Despite there being a significant difference in rectal CM CD8+ T cell frequencies between the vaccine and control groups at the pV#4 and pV#5 time point, when compared with the pre time point, no significant changes were detected in the levels in the vaccinated macaques (Appendix A). Most importantly, the data indicate that the macaques vaccinated with the rSVV-SIV vaccine and protein boosts demonstrated significantly increased mucosal EM CD8+ T cell populations in vaccinated macaques by the end of the fifth vaccination.

### 3.8. CD38+ T cell Activation in Vaccinated Macaques

T cell activation was also indirectly assessed by CD38 surface expression. A significant increase in CD38 expression in vaccinated macaques was evident at post V#4 time points in both jejunum and rectal CD4 and CD8+ T cells compared to controls (*p* <0.001, Figure 6A–D). Pearson’s correlation coefficient analysis between jejunum CD8+CD38+ T cells and the peak plasma VL from control and vaccine macaques indicated a significant negative correlation (r = -0.6612, *p* = 0.014 and r = -0.5903, *p* = 0.026, at pV#4 and pV#5 time points, respectively). Similarly, a negative correlation was also detected between rectal CD8+CD38+ T cell population and the peak plasma VL from control and vaccine macaques, indicating a significant negative correlation (r = -0.5554, *p* = 0.039 and r = -0.5363, *p* = 0.039, at pV#4 and pV#5 time points, respectively). When these changes were compared with their baseline (pre time point) levels, no significant changes in rectal CD4+CD38+ and jejunal and rectal CD8+CD38+ T cell frequencies at pV#5 were observed (Appendix A). We were unable to observe any significant difference in PBMC and LN CD38+ T cells between vaccine and control macaques during any post vaccine time points. Overall, our data suggest that the vaccine induced CD38+ T cell activation in mucosal tissues may be a key regulator in the expansion of mucosal EM T cells as well as in controlling peak plasma VL in vaccinated macaques.

We also measured Ki67 expression in the mucosal T cells during the pre-challenge period and were unable to find any differences between the vaccine and control groups at different vaccination time points (unpublished work). No significant changes were detected in HLADR expression in either CD4 or CD8 T cell populations from rectal or jejunum LPL between pre vaccination and the pV#5 time points (Appendix A). Interestingly, we have observed a significant reduction in CD69 expression in both CD4 and CD8 T cells from jejunum LPL and in CD4 T cells from rectal LPL between pre vaccination and the pV#5 time points (Appendix A). However no significant differences were detected between vaccine and control groups during the pre-challenge times (unpublished work).

### 3.9. rSVV-SIV Vaccination Induced Potent and Polyfunctional Cytokine Responses in Lymph Node Tissue in Vaccinated Macaques

SIV-Gag and -Env-specific CD4+ and CD8+ T cell responses were determined by antigen-stimulated intracellular cytokine flow cytometry staining during the pV#3, pV#4, and pV#5 time points. Three VP macaques (LK50, LK57, and LK58) showed Gag- or Env- specific cytokine responses in LNs throughout the vaccination period (Figure 7A, Table 1 and Appendix A). We observed a greater percentage of Gag-specific cytokine responses (range from 0.0–4.11%) compared to Env-specific responses (0.0–1.91%) in these VP macaques. Interestingly, macaque LK57 was able to generate more Gag- and Env- specific cytokine responses compared to the other two VP macaques. Similarly, these VP macaques generated increased percentages of CD8-specific responses (range from 0.0–4.11%) compared to CD4 responses (range from 0.0–3.66%) responses, although this difference was not statistically significant. All three VP macaques produced CD8+ Env polyfunctional T cell responses for one or two post vaccination time points while Gag-specific polyfunctional responses were limited to LK57 only (Figure 7B and Appendix A). LK57 also induced both Env-specific polyfunctional CD4 and CD8 T cells (Figure 7B). Interestingly, LK58 had shown the highest polyfunctional responses (4–5 cytokines) at pV#4, however, there were no polyfunctional responses detected at the pV#5 time point. The reason for the absence of polyfunctional responses at pV#5 in LK58 is not obvious and will require additional study.

To determine whether the antigen-specific responses between VP (LK50, LK57, and LK58) and VI macaques (LK45, LK54, LK56, LK59, and LK60) had any impact in deciding the outcome of the challenge study, we also quantified SIV antigen-specific responses in all VI subjects. Similar to VP macaques, all VI macaques also generated Gag- and Env- specific responses for one to three post vaccination time points (Figure 7C and Table 1). Gag- or Env- specific CD4 responses were detected at only one time point in LK45 and LK59 VI animals (Appendix A). The other VI macaques also showed similar Gag (range from 0.0–2.46%)- and Env (range from 0.0–8.00%)- specific cytokine responses (Appendix A). LK56 had the highest cytokine producing responses compared to the rest of the VI macaques, and showed responses at all the time points tested (Figure 7C). More importantly, LK56 demonstrated the highest polyfunctional CD4 and CD8 responses (6 cytokines) at the pV#3 time point, however, this increase was not maintained in the subsequent time points (Figure 7B). Overall, only three out of five VI macaques that did not resist challenge and became SIV+ had detectable cytokine responses at the pV#5 time points, whereas all the 3 VP macaques had cytokine responses at the pV#5 time point in peripheral LN (Figure 7 and Table 1).

SIV-specific responses in PBMC were similar in both VP (Gag: range from 0.0–4.74%, Env: range from 0.0–4.39%) and VI (Gag: range from 0.0–4.37%, Env: range from 0.0- 4.35%) macaques in all the pre-challenge time points tested (Table 1 and Appendix A, Appendix A). Two out of three VP macaques had detectable polyfunctional Gag- or Env- specific responses (maximum 3 cytokines) at the pV#3 or pV#5 time points (Appendix A). All VI macaques demonstrated polyfunctional Gag- or Env- specific responses (1–4 cytokines) at various post vaccination time points (Appendix A).

### 3.10. Vaccination Induces Antigen Specific Cytokine Responses in Jejunum

Gag-specific cytokine responses in LK50, LK57 and LK58 at pV#3, Gag-specific (LK58) and Env-specific (LK57 and LK58) responses at pV#4, and only Gag-specific (LK57) responses at pV#5 were measured in VP macaques. Both LK57 and LK58 induced strong Gag (range from 0.0–7.99%)- and Env (range from 0.0–5.0%)- specific cytokine responses (Figure 8A,B). These two VP macaques had more cells expressing a combination of two or three cytokines in response to Gag or Env antigens than those detected in VI macaques (Figure 8C, Appendix A). We were unable to perform mucosal antigen-specific cytokine assays in all vaccinated animals using both Gag and Env antigens due to the limitation of available cells. However, based on our limited tissues and antigen-specific assays, the VI macaques were shown to be unable to produce polyfunctional cytokine responses in the gut and out of the detectable responses, most were monofunctional (Gag: responses range from 0.0–2.41%; Env: range from 0.0–1.73%) (Appendix A), Hence, VP LK57 and LK58 demonstrated both CD4+ and CD8+ polyfunctional cells with expression of two or three cytokines in response to Gag or Env antigens in the jejunum LPL compared with the monofunctional cells demonstrated in the VI macaques.

## 4. Discussion

SIV infection in rhesus macaques strongly recapitulates HIV infection in humans, and is the most evolved animal model available to test candidate SIV vaccines that have the capability of later translation to HIV vaccines [46]. The vaccine candidates we tested utilized a live attenuated SVV vector backbone to deliver SIV *env* and *gag* genes. Our earlier studies demonstrated that SVV vectored SIV*env* and SIV*gag* vaccines are immunogenic, generate neutralizing and cellular immune responses to SIV, findings that are likely responsible for the significant 1.5 log reduction of circulating SIV viral loads in the vaccine animals [24,25,26].

The hypothesis for this present vaccine study was that the use of both intranasal and subcutaneous rSVV-SIVenv and rSVV-SIVgag immunizations to increase mucosal immune responses, the added SIV gp120 Env and p55 Gag proteins with non-alum adjuvanted intramuscular boosts, and the usage of repeated mucosal intravaginal challenges with uncloned SIVmac251 virus stock, would increase SIV-specific immune responses, provide a more natural route and dose for SIV challenge, and produce greater vaccine efficacy. The multiple immunization time points proposed are validated by our earlier findings that showed that prior immunity to SVV does not diminish immune responses to the vaccine (unpublished work). This study showed that the vaccine was able to protect three out of eight (37.5%) vaccinated animals against pathogenic SIV challenge when compared to controls. These observations confirm previous findings that the rSVV-SIV vaccines are fully attenuated in RMs, likely due to the insertional inactivation of the SVV glycoprotein C gene, a possible varicella virulence gene [47,48]. Although not confirmed in this study, defective gC expression of the VZV Oka vaccine had been documented earlier [48,49]. Although five vaccinated macaques did have breakthrough SIV infection, these VI RMs showed a significantly greater (>2 log) VL reduction from controls, compared with the 1.5 log VL reduction that was documented in our earlier rSVV-SIV vaccine study [25]. Our results are similar to the modest efficacy of 31.2% also seen in the first successful HIV vaccine trial, Thai RV144 [50]. The RV144 vaccinations included IM injections of the ALVAC canarypox vector expressing HIV*env* and HIV*gag* and with two IM booster injections of alum-adjuvanted HIV gp120. The RV144 vaccine regimen induced mainly non-neutralizing Abs and CD4+ responses, with Abs directed to the V1/V2 of gp120 found to be a primary correlate of reduced risk of HIV acquisition [42,51].

The rSVV-SIV vaccine regime in the current study induced robust Gag- and Env-specific IgG titers along with SIV specific CD4+ T cell responses, suggesting a contribution of CD4+ T cell help in the generation of vaccine specific antibody responses. High IC50 levels of neutralizing antibodies were detected in all vaccinated macaques at the pre challenge time point. However, two of the three VP macaques, LK50 and LK57, generated much higher IC50s than the other VI macaques in this study or those rSVV-SIV only vaccinated macaques in our earlier study that also did not produce protection [25], which suggests the SIV-specific Nabs may have contributed to their partial protection. The weak Nab response detected in one VP animal (LK58) pre challenge further suggests that vaccine induced protection can be more complex and that other factors are involved. After challenge and one month after infection, the VI macaques (LK45, LK54, LK56, LK60) also generated high Nab IC50s similar to the two VP macaques at 58 wks post immunization. This high Nab response in VI macaques following SIV infection suggests that the high Nab titers were not capable of neutralizing completely the replicating pathogenic viruses. It is likely that the high levels of Nabs observed are a response from the neutralizing-sensitive viruses present in the challenge swarm virus population and not broadly the Nabs, as those have been shown to develop very late after infection [39]. VP macaques at this later time point showed a decrease Nab response with no further SIV antigen stimulation.

Immediate increased SIVGag-specific IgG responses at the early pV#3 time point in two VP macaques, LK50 and LK57, was also suggestive of early production of non-neutralizing binding antibodies that may also correlate with protection or reduced risk of infection. LK58, the third VP RM, also maintained strong non-neutralizing SIVGag- and Env- specific IgG responses during the challenge phase with minimal neutralizing antibody titers, which suggests a potential role of non-neutralizing antibodies in providing protection from pathogenic challenge.

Findings of robust non-neutralizing binding antibodies to SIVGag, SIVEnv, and SIV gp70 V1/V2 antigens in the vaginal secretions of the vaccinated RMs prior to SIV challenge demonstrated the ability of this vaccine to induce early mucosal immune responses in the target vaginal tissue. All VI SIV+ vaccinated RMs generated robust peripheral SIV specific IgG responses at the time of pathogenic challenge. Our study was unable to predict the outcome of mucosal SIV challenge based on peripheral and limited mucosal binding antibody responses due to the limited number of VP macaques. Additionally, the relative contributions of the rSVV-SIV presented SIV gp120 and p55 genes versus the SIV Env and Gag protein boosts are not clear. This warrants future studies to include vaccination with single as well as a combination of different genes/antigens, including the rSVV-*gag*, rSVV-*env* genes, gp120 protein, p55 protein, with or without adjuplex adjuvant.

Long-term immunological protection from infection depends on increased differentiation of effector memory T cells [52]. Increased memory T cell inflation has also been reported to protect from pathogenic CMV challenge [53,54]. In the current study, vaccination induced increased EM CD4 population in peripheral LN during the pV#3 and pV#4 time points, which is suggestive of the conversion of naïve cells to memory T cell populations. Moreover, increased differentiation to effector CD4+ T cells would help to increase B cell responses as detected by increased peripheral antigen-specific humoral immune responses in all vaccinated macaques. The majority of mucosal CD4 and CD8 cells are activated memory cells and can produce cytokines in the presence of an antigen stimulant [55,56]. A significant increase of EM CD8+ T cells and a converse decrease of CM CD8+ T cells were documented and suggest enhanced differentiation of EM CD8+ T cells in all vaccinated macaques. Increased CD38 expression in mucosal CD8 populations also supports our increased differentiation of CD8+ T cells, suggesting that this vaccine was able to generate EM CD4/CD8 differentiation and activation. Absence of any changes in the peripheral EM T cell population in vaccinated macaques suggests that the vaccine induced changes were predominantly present in mucosal and lymphoid tissues. Despite a significantly greater reduction in plasma VL in the vaccinated macaques, the vaccine was unable to control the initial destruction of intestinal CD4+ T cells in VI RMs in the short 12 wks post infection follow-up period. However, it is important to remember that the initial intestinal CD4+ T cell loss has also been detected in natural non-progressor macaques [57,58,59], suggesting that the initial CD4 loss does not predict the long-term outcome of the vaccine in our study. We detected a significant reduction in the peak plasma VL as well as in the maintenance of lower plasma VL in vaccinated macaques when compared to controls, and postulate that those VI macaques might become long-term non-progressors with longer than 12 wks follow-up. This will require future study.

We conclude that vaccine mediated mucosal polyfunctional responses contribute to protection in the three VP macaques compared to the five VI macaques where monofunctional cytokine responses were predominant. These findings support that the quality of cytokine responses are critical to determine the outcome of an SIV vaccine [60,61]. We did not observe any difference in the peripheral blood polyfunctional cytokine responses between the VP and VI macaques, which indicates that the vaccine mediated protection with this vaccine regimen is more mucosal and lymphoid tissue centric.

## 5. Conclusions

Overall, our data suggest that detectable and qualitative peripheral and mucosal IgG responses to SIVGag, SIVEnv, and SIV V1/V2 binding antibodies and Nabs, in addition to mucosal and lymphoid polyfunctional cytokine responses, are crucial to mediating protection from infection in three out of eight vaccinated macaques, showing a 37.5% efficacy, and a significant reduction in plasma viremia in the remaining five vaccinated RMs. These data support the further investigation of rSVV-SIV as a potential vaccine candidate. Ultimately, a recombinant VZV vaccine expressing HIV antigens may produce the efficacy needed to combat the ongoing AIDS epidemic.

## Figures and Tables

**Figure 1 viruses-14-02819-f001:**
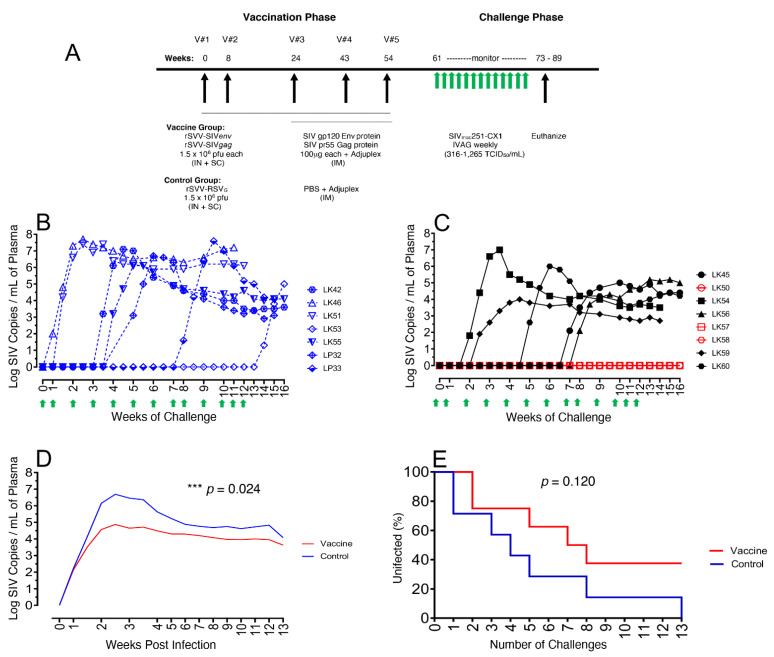
Study design and post infection SIV plasma viral loads. (**A**) Two groups of macaques (RMs) were immunized with either rSVV-SIVenv/gag (vaccine group, *n* = 8) expressing *gag* and *env* of the SIVmac239 clone and/or boosted with the native form of SIVmac251/p27 Gag and gp120 Env adjuvanted in 5mg of Adjuplex) or rSVV-RSVg (control group, *n* = 7) expressing the G protein of RSV and/or given a mock boost of Adjuplex and PBS. At 61 weeks post initial immunization, the challenge phase began with administration of up to 13 repeated weekly intravaginal challenges with the pathogenic, SIVmac251-CX-1 until all control macaques were infected. Twelve weeks post SIV infection or at the end of the study, all macaques were euthanized. Logarithmic longitudinal plasma viral loads (VL) determined by RT qPCR SIV copies per milliliter plasma following intravaginal challenge for each of the 13 multiple intravaginal challenges and showing the number of challenges required for infection of each of the control with dashed blue lines (**B**), vaccine macaques with black solid lines for vaccinated and infected (VI), and red solid lines for vaccinated protected (VP) during challenge phase (**C**). Each intravaginal SIV challenge is shown by upward green arrow (**A–C**). (**D**) Mean log longitudinal plasma VL for both vaccine (red) and control (blue) macaques post SIV infection showing a significant reduction of the VL in vaccine macaques compared to controls (*p* = 0.024). (**E**) Infection curves following multiple intravaginal SIVmac251-CX-1 challenges for vaccine (red) and control (blue) immunized rhesus macaques are shown.

**Figure 2 viruses-14-02819-f002:**
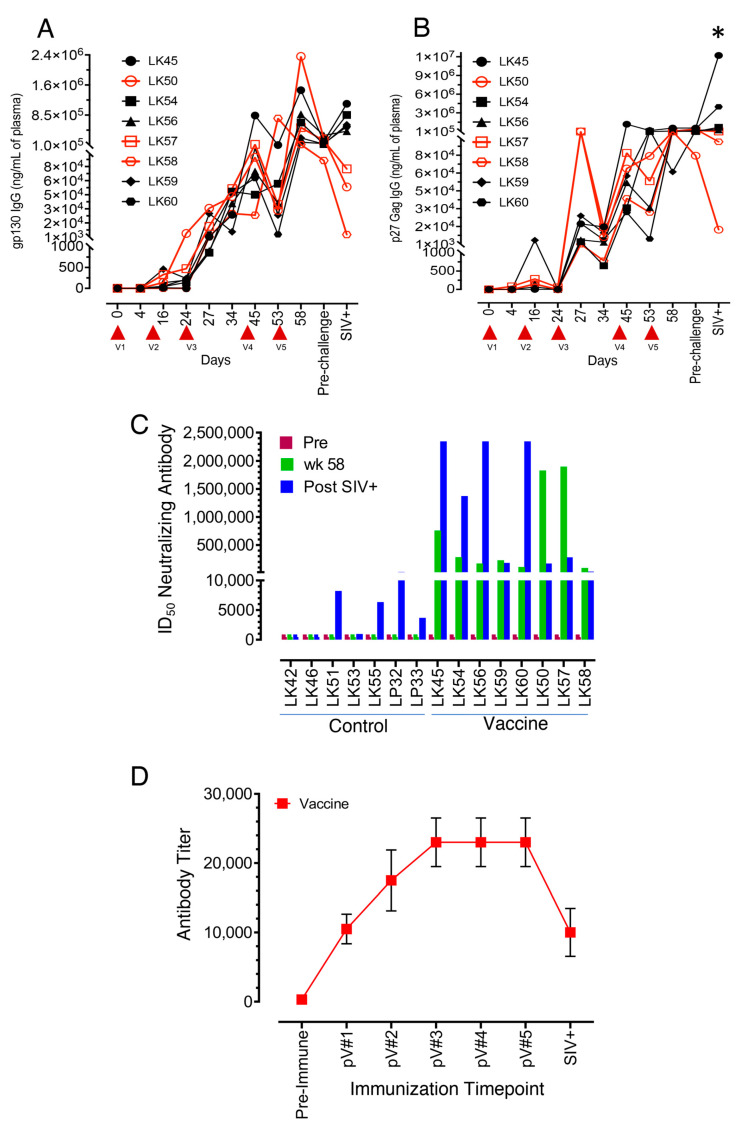
Humoral antibody responses in vaccine macaques. (**A**,**B**) Antigen-specific IgG responses measured over the vaccination and post challenge time points in all vaccinated macaques (*n* = 8). Individual SIVgp130 (**A**) and SIVp27 (**B**) specific IgG responses are shown on the Y axis. Note that all vaccinated macaques had increased anamnestic IgG responses detected after second boost (V3) immunization. Red arrows on the X-axis denote vaccination (V) time points. Red and black lined data represent vaccine protected (VP) and vaccine infected (VI) macaques, respectively. X axis denotes weeks post immunization, pre-challenge, and 1 month after SIV confirmed infection time points. Asterisk denotes statistically significant differences in SIVGag specific IgG responses between 1-month SIV+ and pre vaccination time points (* *p* < 0.05). (**C**) SIV-specific neutralizing antibodies were measured over the vaccination and post challenge time points in all control (left) and vaccine (right) macaques at pre and post vaccine #5 at 58 weeks post initial immunization and prior to SIV challenge, and 1 month post SIV challenge. Following vaccine #5 at week 58, neutralizing antibodies were induced in all vaccine macaques with significant increases in LK50 and LK57 (*p* = 0.012). Vaccinated macaques LK45, LK54, LK56, LK59, LK60 became SIV infected, while vaccinated macaques LK50, LK57 and LK58 remained SIV negative through 13 challenges. The ID_50_ neutralizing antibody is the reciprocal serum dilution, reducing RLU by 50% in TZM-bl cells compared to virus control. (**D**) Mean SVV vector-specific IgG titers were measured over the vaccination and post challenge time points in all vaccinated macaques (*n* = 8). SVV antibody titers were expressed as the reciprocal of the serum dilution that resulted in A_450_ absorbance readings above the determined cut-off value. All macaques had increased anamnestic responses detected after vaccines #2 and #3 but plateaued for vaccines #4 and #5. SVV Vector antibody levels dropped off following SIV challenge and/or infection.

**Figure 3 viruses-14-02819-f003:**
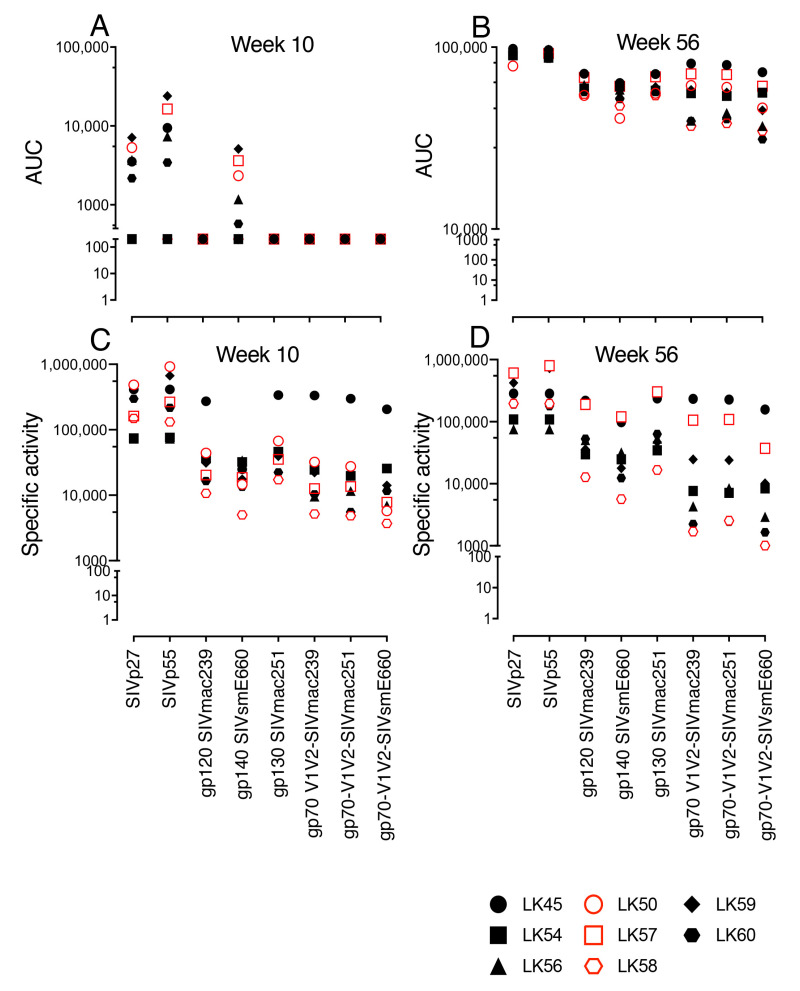
BAMA assay. SIV-specific binding antibody (BAMA) data from all vaccine macaques (*n* = 7-8) at 10 weeks and 56 weeks post initial immunization in serum (**A**,**B**) and vaginal secretions (**C**,**D**). Individual SIV antigens tested are shown in X axis. Serum IgG binding magnitude data are shown as area under the curve (AUC) in the Y axis (**A**,**B**) while vaginal secretions data are shown as Specific Activity in the Y axis. (**C**,**D**). Vaccinated protected (VP) and vaccinated infected (VI) RMs are shown with red and black symbols, respectively.

**Figure 4 viruses-14-02819-f004:**
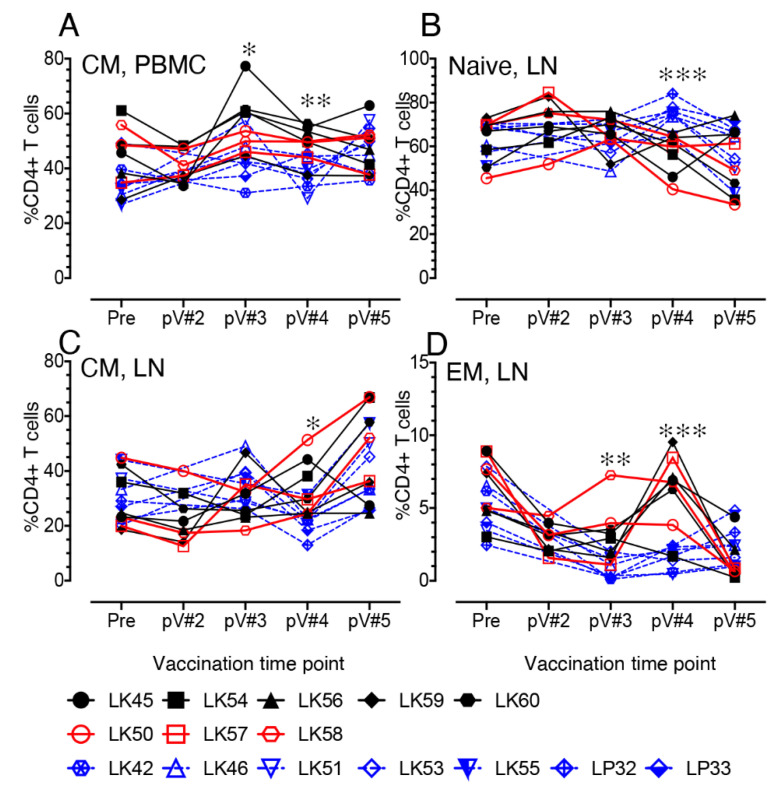
Dynamics of different T cell population during vaccination in vaccinated and control macaques in PBMC and lymph node tissues. Frequency of central memory (CM, **A**,**C**), naïve (**B**), and effector memory (EM, **D**) CD4 T cell population in PBMC and lymph node (LN) is shown for vaccine (*n* = 8) and control (*n* = 7) animals over the vaccination time points. Black and red lines represent vaccine infected (VI) and vaccine protected (VP) macaques, respectively. Blue dashed lines represent control macaques. Asterisks denote statistically significant differences between vaccine and control groups for the specific time point (* *p* < 0.05, ** *p* < 0.01, and *** *p* < 0.001).

**Figure 5 viruses-14-02819-f005:**
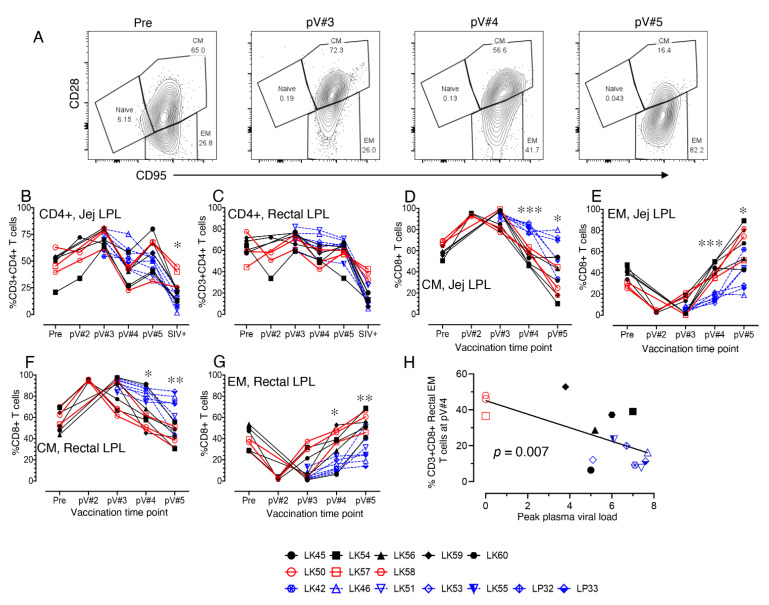
Dynamics of naïve and memory T cell populations during vaccination and post SIV infection in vaccinated and control macaques in mucosal tissues. (**A**) Representative contour plots showing the increased population of effector memory (EM) rectal lamina propria lymphocytes (LPL) CD8+ T cells in a macaque LK50 over the course of vaccination. Frequency of jejunum (**B**) and rectal (**C**) CD4+ T cells are shown for both vaccine and control macaques. A rapid loss of mucosal CD4+ T cells is detected in all infected macaques one month post confirmed infection. There was no loss of mucosal CD4+ T cells detected in vaccine protected (VP) macaques. Dynamics of central memory (CM; **D**,**F**) and EM (**E**,**G**) CD8 T cell population in jejunum and rectal LPL are shown for vaccine (*n* = 8) and control (*n* = 7) macaques over the vaccination time points. Interestingly, all vaccinated macaques had a significantly higher effector memory CD8 T cell detected from 2 weeks post vaccination 4 (pV#4) onwards. (**H**) Pearson’s correlation analysis demonstrated a significant negative correlation between rectal EM CD8+ T cell frequencies at pV#4 and peak plasma VL between vaccine and control macaques (*p* = 0.007, r = -0.6637). Black and red lines represent vaccinated infected (VI) and VP macaques, respectively. Blue dashed lines represent control macaques. Asterisks denote statistically significant differences between vaccine and control groups for the specific time point (* *p* < 0.05, ** *p* < 0.01, and *** *p* < 0.001).

**Figure 6 viruses-14-02819-f006:**
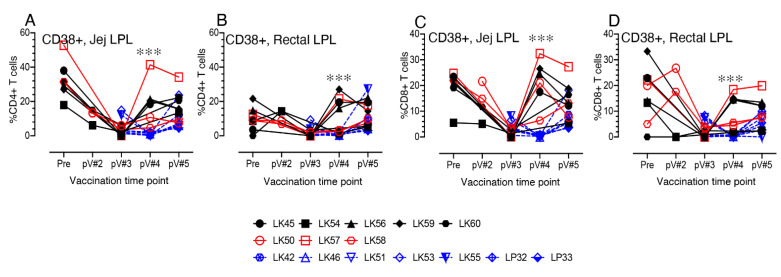
Dynamics of CD38+ activated T cell populations during vaccination and post SIV infection in vaccinated and control macaques in mucosal tissues. Frequency of activating CD4+ (**A**,**B**) and CD8+ (**C**,**D**) jejunal and rectal T cells are shown for both vaccinated and control macaques. A significant upregulation of activating CD4 and CD8 T cells are detected in vaccinated animals compared to control animals at 2 weeks post vaccination 4 (pV#4) time point. Black and red lines represent vaccinated animals. Black and red lines represent vaccinated infected (VI) and vaccine protected (VP) macaques, respectively. Blue dashed lines represent control macaques. Asterisks denote statistically significant differences between vaccine and control groups for the specific time point (*** *p* < 0.001).

**Figure 7 viruses-14-02819-f007:**
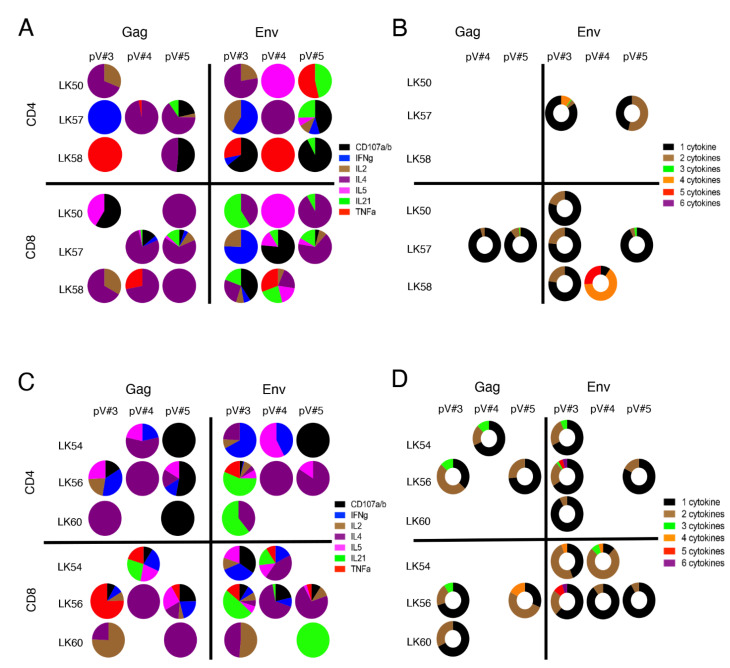
Polyfunctional CD4+ and CD8+ T cell response to vaccine antigens in vaccinated protected (VP) and vaccine infected (VI) macaques. Peripheral LNs were isolated at the two weeks post vaccination (pV) time points, including pV#3, pV#4, and pV#5, and were stimulated with overlapping Gag and Env peptide pools. Following stimulation, cells were stained for the presence of CD107a/b, IFNγ, IL2, IL4, IL21, and TNFα. Each segment of the pie represents the proportion of different cytokines produced by either CD4 or CD8 T cells in presence of SIV antigens in VP (**A**) and VI (**C**) macaques. Each segment of the donut represents the proportion of CD4+ or CD8+ T cells secreting the combination of multiple cytokines in VP (**B**) and VI (**D**) macaques.

**Figure 8 viruses-14-02819-f008:**
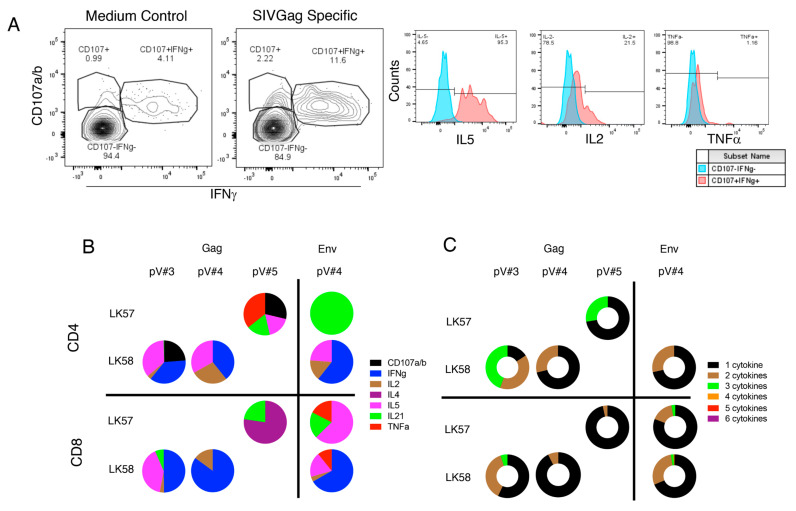
Polyfunctional CD4+ and CD8+ T cell response to vaccine antigens in jejunum lymphocytes from vaccinated protected (VP) macaques. Jejunum lamina propria lymphocytes (LPL) were isolated at two weeks post vaccination (pV) time points including pV#3, pV#4, and pV#5, and were stimulated with overlapping Gag and Env peptide pools. Following stimulation, cells were stained for the presence of CD107a/b, IFNγ, IL2, IL4, IL21 and TNFα. (**A**) Representative contour plots of jejunum LPL CD4+ T cells showing CD107a/b and IFNγ expression in presence of SIVGag peptides compared to medium control from a vaccine protected LK58 macaque. The frequency of each subset is shown in the box of each gated plots. The jejunum CD107a/b- IFNγ- CD4+ T cells (blue histograms) and CD107a/b+ IFNγ+ CD4+ T cells (red histograms) were further analyzed for expression of IL5, IL2 and TNFα. Percentages of positive and negative population for each cytokine for CD107a/b+ IFNγ+ CD4+ T cells are shown at the top of each histogram. (**B**) Each segment of the pie represents the proportion of different cytokines produced by either CD4 or CD8 T cells in presence of SIV antigens. (**C**) Each segment of the donut represents the proportion of CD4+ or CD8+ T cells secreting the combination of multiple cytokines.

**Table 1 viruses-14-02819-t001:** SIV antigen-specific cytokine responses in LN and PBMC from vaccinated macaques at pV#5 time point.

Macaque	Antigen #	Tissue	% of CD3^+^CD4^+^ T cells							% of CD3^+^CD8^+^ T cells						
			CD107a/b	IFNγ	IL2	IL4	IL5	IL21	TNFα	CD107a/b	IFNγ	IL2	IL4	IL5	IL21	TNFα
LK50	Gag	LN											2.99			
LK57	Gag	LN	0.53		0.09	1.62		0.24		0.27	0.22	0.54	3.62	0.15	0.74	
LK58	Gag	LN	0.36		0.34								1.04			
LK54	Gag	LN	0.48													
LK56	Gag	LN	1.75	0.44		0.57	0.54			1.49	1.26	0.32	0.95	1.49		0.50
LK60	Gag	LN				0.63							0.99			
LK50	Env	LN						0.19	0.22				1.61		0.14	
LK57	Env	LN	0.52	0.12	0.12		0.09	0.29		0.12		0.22	1.91	0.17	0.48	
LK58	Env	LN	0.54					0.05								
LK54	Env	LN	0.52												0.10	
LK56	Env	LN				0.74	0.14			0.33		0.37	2.55	0.11		0.17
LK50	Gag	PBMC	0.06										0.45	0.15		
LK58	Gag	PBMC					0.05									
LK45	Gag	PBMC	0.18			1.58							4.37			
LK54	Gag	PBMC	0.45													
LK56	Gag	PBMC	0.51										3.14			
LK59	Gag	PBMC	0.09	1.50	0.11	1.73	1.12	0.46	0.33		0.65		3.44	0.75		
LK60	Gag	PBMC	0.24							0.17						
LK50	Env	PBMC											0.52	0.17		
LK57	Env	PBMC		1.41	0.16			0.11			0.54					
LK58	Env	PBMC											1.54			
LK45	Env	PBMC	0.50													
LK54	Env	PBMC	0.32							1.22			1.12			
LK59	Env	PBMC	0.30	0.38		1.30	0.25	0.35			0.37		1.96	0.37	0.35	
LK60	Env	PBMC	0.09							0.08				0.09		

# Peptide-specific responses are shown as percentage for CD3+CD4+ and CD3+CD8+ T cell subsets. Although both Gag and Env peptides pools were tested for all vaccinated macaques, only positive responses are shown. Note that the red lettered macaques were vaccinated and protected after repeated challenges.

## Data Availability

All relevant data are included within the manuscript. The raw data are available on request from the corresponding authors.

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
