# Peer review of "Recombinant Simian Varicella Virus-Simian Immunodeficiency Virus Vaccine Induces T and B Cell Functions and Provides Partial Protection against Repeated Mucosal SIV Challenges in Rhesus Macaques"

_viruses, 2022, doi:10.3390/v14122819_

Round 1

Reviewer 1 Report

This manuscript describes the production of a recombinant simian varicella virus containing inserts from SIV, with the goal to create a SIV vaccine. Three of 8 vaccinated monkeys remained uninfected after repeated challenges with SIV. The manuscript is well written and well documented. Even though the authors may have desired even higher effectiveness, the vaccine data contained in this manuscript will be helpful in the preparation of next generation vaccines against SIV and ultimately HIV. The use of monkeys as an animal model for human vaccines is encouraged.

1.Methods. Animal cohort, line 145. Explain why only female monkeys were used. Put answer in manuscript. 

2.Results and supporting Information. Lines 1064-1103. Please prepare a small Table within Results, to summarize the Cytokine expression results currently shown only in Supplemental Figures 2-8. List each of the 6 cytokines in the Table. The authors can still keep the supporting figures. 

3. Discussion. Add new information. In the Introduction, the authors briefly mention the use of a recombinant varicella virus with inserts from other viruses. Please provide a new short paragraph in the Discussion to describe these interesting vaccine studies by Yamanishi and Mori. For example, see article entitled “Novel polyvalent live vaccine against VZV and mumps virus, “2013. Describe what was learned from these recombinant varicella vaccine candidates

Reviewer 2 Report

Pahar et al

To the authors: The authors describe a simian varicella driven SIV vaccine (SVV) candidate tested in rhesus macaques with subsequent repeated intravaginal challenge. This approach appears to be under-researched, given the limited host range of the vector precluding small animal models. 5 varicella vaccine shots are supplemented by gag and env protein boosts supplementing shots 3-5. A control group receiv ed SVV expressing RSV G in 5 shots, supplemented by adjuvant only at shots 3-5. The group followed SIV infection, binding antibodies, neutralizing antibodies, anti-vector antibodies, antibodies to different forms of Gag and Env including V1V2 scaffolds, total T cell responses of different compartments and function in PBMC and tissues, antigen-specific cytokine secreting T cell responses again in PBMCs and mucosal tissues/lymph nodes. 

This impressive paper is clearly the product of years of planning, a huge amount of diligent no stone unturned effort on a spectrum of immune responses. Below are a number of comments and suggestions in no particular order of priority:

The binding antibody tests suggest that the varicella vaccine does not induce much Ab responses to either env or gag much above baseline, but may prime B cells ahead of the protein boosts. This justifies the protein boosts, but isn’t actually mentioned in the paper but could be to help the reader understand the decision making. 

Regarding the neutralization assay titers being 10E6 in some vaccinees, this seems to be incompatible with the concept of a tier 2 challenge virus, at least if the term implies a similar phenotype to HIV strains. The highest autologous tier 2 HIV NAB titers I have seen published in vaccinated animals is about 1,000-10,000, so the titers in the current paper is at least a couple of orders of magnitude higher than that. These ID50s may be similar their high binding titers, which implies that the 251-CX1 strain used in neutralization and for challenge is more like a tier 1 virus, not a tier 2 virus as claimed. Based on published HIV antibody protection studies, titers of ~1:200 would be expected to protect, so these titers are very much higher, which makes it unclear why protection was not more robust. Also, the nature of the gp120 protein boost (i.e., not resembling a native tier 2 trimer) would not lend itself to tier 2 neutralization titers, as in the RV144 human trial and elsewhere. Since gp120 accounts for the major boosting of antibody responses in RMs, one would not expect such high tier 2 responses. If this is a tier 2 virus, does gp120 immunization alone also induce very high autologous neutralization titers? This would be important to know, because it impacts the value of this challenge for human HIV infection. The potential use of a SHIV model for this kind of work might be helpful for vaccine development, although this has other drawbacks. Taking a different view of potentially protective antibodies, the measurement of V1V2 responses suggests the authors may be expecting non-neutralizing antibodies (at least to tier 2 viruses) that may still mediate a protective Fc mediated cellular response. 

The correlates of protection are not entirely clear: they could be CD8+ EM, based on Fig 5H and or polyfunctional cytokine secreting T cells in mucosa (Fig. 7) or higher nAbs in some VP animals 50 and 57. The fact that soluble proteins were used as well as the SVV vector in many shots but with no animals getting just one or the other makes it difficult to gauge the relative contributions. While the sum of the two vaccines may be more powerful than one, a protein boost group would have been better lacking the RSV SVV prime.

Line 85: the 2012 paper says that increased cellular immune response and CD4+ T cell proliferation correlate with reduced plasma viral load, hence, "negative correlated" is wrong, should be correlated?

Lines 334-340 could be put either in methods or supporting information so that the results jumps straight into the immune responses and challenge data. Could say immunizations were well tolerated and refer to methods? 

Fig 1B & C in legend does not coincide with the figure. In Figure 1B, there are 7 animals and 1C has 8 animals. Hence B should be "Control" and C should be "Vaccine" in the figure legend OR swap the figure arrangement in Figure 1.

Also, inconsistent color for "vaccine" group, whereby there are black and red symbols for protected and unprotected.  Need to define the colors (black for infected during challenge, red for uninfected during challenge). Also red line can be confused with red arrows in Fig 1B & 1C. I know red arrows indicate challenge weeks, but it would be good to use another colour to differentiate things.

It would be good to know if prior immunity to the SVV affects its use for later vaccine delivery. For example, by preceding SVV vaccine doses with two shots of the control (RSV) vector. The vector immunity figure only aprtly addresses this point in detecting immunity but not how the immunity might dampen vaccine responses, or not? 

How is the env construct made? Is it full length gp160 or gp120, a mutant? Does it make trimers? 

line 146 is SRV meant to be RSV?

Line 453 Figure 4. The analysis appears to detect total CD4 phenotypes (i.e., not antigen-specific). Given that both control and vaccine groups received rSVV vectors (SIV or RSV) throughout, can we infer that differences between the vaccine and control groups pertain to the presence or absence of protein boosts given in Adjuplex in shots 3-5? This seems to be the case, reading the sentence starting on line 467 on CM, which perhaps should be highlighted more as the difference could reflect the key difference in regimens between groups. 

Figure 5B, C. Should the y axis be CD4+ not CD3+? Or double positive?

Figure 7. The polyfunctional responses show a wide variety of cytokines at different timepoints and between animals. I don’t know what to make of this. The text on line 542 and beyond talks about % positives, but there is no figure on the %. It would be good to have a sentence to sum up the cytokine secreting work at line 580. There is a lot of information in this data and much work went into it, so a “10,000 foot” view of it would be good to tie it together. It might be useful to try to summarize it in a figure? This is covered by the final paragraph of the discussion pointing to mucosal protection, but it could be covered in the results to not leave the point hanging for the reader after so many figures. 

Fig 8C only shows animals 57 and 58, so line 587 comparing to VI macaques is not really shown in the figure. Might the reduced antigen specific responses in other (infected) animals relate to the lower cell numbers in these animals (Fig 5B, C?). 

Round 2

Reviewer 2 Report

I looked at the comments.

 Responses to all points are OK, but I still have reservations about #5

 I still find it very unusual that a vectored gp160 prime with Gag and gp120 protein boost would give such high nAbs to a tier 2 virus, even if matched to the strain. Even the unprotected vaccinated animals raise huge titers of about 1:200,000, the protected ones being only a log higher.

 The authors say the two VP animals had much higher tier 2 nAb titers than expected, like SIV infected macaques. However, I would say even the VI macaques have much higher titers than expected based on what I have seen for HIV.

 The extremely high tier 2 titers in infected macaques may be a clue here. Such high tier 2 autologous nAb titers are rare in HIV+ infection, again being a couple of orders of magnitude lower.

 Overall, I think the author’s response goes only part way to addressing the issue only as it points only to the very high titers in the VP animals. I think it needs to cover the point that ALL the animal titers are extremely high in comparison to what is seen in HIV. Also the post-infection titers are unusually high, as if everything is 2-3 logs higher than typically seen in the neut assay. As mentioned in my initial comment, if we take RV144 as an imperfect comparator for HIV, involving a pox vector prime with gp120 boost, this vaccine induced undetectable nAb in humans. Whereas here we see a superficially similar vaccine involving a vector prime and gp120 boost that induces radically higher titers of 10E5-10E6.

I would really like to know what makes such a difference:

Why would nAb titers of 10E5 not protect when basically 10E2 nAbs protect against HIV?

What explains the radically different titers in all samples? Is it something about the vector prime? The fact the vector prime is co-immunized with boost unlike RV144? Is it something about the challenge virus that somehow makes it sensitive to vaccine nAbs, despite its characterization as a tier 2 virus? Is there something special about macaque antibodies that allows them to neutralize so robustly, unlike human antibodies produced by equivalent HIV immunogens?

 In the methods, it mentions that SIV viruses were mac251 CX1 live stocks or SIV pseudoviruses. Which was it? If live virus was used, is it possible that passaging led the virus to become relatively sensitive?

 In the Welles et al 2022 paper I could not find the 251-CX-1 or the -PM (previous name) among the viruses listed by a word search? Despite the claim that the paper reveals it to be a tier 2 virus. In Fig. 1 two unusually potent plasmas from infected NHPs are analyzed with high NAB titers. The first only gave titers of up to 2400 ID50 against tier 2 viruses and only slightly higher against some tier 1 SIV strains. The second plasma gave a very high autologous tier against a tier 2 matched strain and a tier 1 virus of 10E6. Thus, we can infer that high autologous titers in the 10E6 range to tier 2 virus is very rarely possible, but this was due to infection, not a vaccine as in the present paper. However, there is nothing I could find in this paper to support the tier 2 status of CX-1. The point also remains that it is really tough to get past the observations that an ID50 of 10E5 is insufficient to protect but 10E6 is enough.

 Given the similarity of the binding and neut titers, the CX1 used in the neut assays appears to be highly sensitive, but the lack of protection across the board protection suggests that the challenge virus may be more resistant. Were the CX-1 used in NAB assays the exact same live stocks?

 Given these concerns, it might be useful to directly cross-reference the NAB assay/virus used in this paper with some of the MABs or neutralizing plasmas in the Welles 2022 paper to make sure that the sensitivity of the assays is reasonably equivalent.  
